# Ultra-Thin SnS_2_-Pt Nanocatalyst for Efficient Hydrogen Evolution Reaction

**DOI:** 10.3390/nano10122337

**Published:** 2020-11-25

**Authors:** Yanying Yu, Jie Xu, Jianwei Zhang, Fan Li, Jiantao Fu, Chao Li, Cuihua An

**Affiliations:** Center for Electron Microscopy, TUT-FEI Joint Laboratory, Tianjin Key Laboratory of Advanced Porous Functional Materials, Institute for New Energy Materials & Low-Carbon Technologies, School of Materials Science and Engineering, Tianjin University of Technology, Tianjin 300384, China; yuyanying_link@163.com (Y.Y.); xujielink@163.com (J.X.); zhangjianwei248@163.com (J.Z.); lifan_96@163.com (F.L.); fujt2020@163.com (J.F.)

**Keywords:** ultra-thin SnS_2_ nanocatalyst, Pt nanoclusters, synergistic effect, hydrogen evolution reaction

## Abstract

Transition-metal dichalcogenides (TMDs) materials have attracted much attention for hydrogen evolution reaction (HER) as a new catalyst, but they still have challenges in poor stability and high reaction over-potential. In this study, ultra-thin SnS_2_ nanocatalysts were synthesized by simple hydrothermal method, and low load of Pt was added to form stable SnS_2_-Pt-3 (the content of platinum is 0.5 wt %). The synergistic effect between ultra-thin SnS_2_ rich in active sites and individual dispersed Pt nanoclusters can significantly reduce the reaction barrier and further accelerate HER reaction kinetics. Hence, SnS_2_-Pt-3 exhibits a low overpotential of 210 mV at the current density of 10 mA cm^−2^. It is worth noting that SnS_2_-Pt-3 has a small Tafel slope (126 mV dec^−1^) in 0.5 M H_2_SO_4_, as well as stability. This work provides a new option for the application of TMDs materials in efficient hydrogen evolution reaction. Moreover, this method can be easily extended to other catalysts with desired two-dimensional materials.

## 1. Introduction

Hydrogen is a considerable chemical commodity for its application in ammonia synthesis and petroleum refining [1,2,3,4]. Electrocatalysts for hydrogen evolution from water have been extensively studied for their advantages, having high purity and use in environmentally friendly products [5,6]. Now, Pt [7] and Pt-based catalysts have been considered as the most effective HER electrocatalysts reported in the literature [8]. However, the poor electrochemical stability, high production costs [9] and the raw material scarcity limits their mass production to meet the industrial demand. Therefore, the design HER catalyst of economic efficiency has become the key factor of electrocatalytic water splitting.

In recent years, the electrocatalytic properties of transition metal sulfides [10,11], carbides [12,13,14], borides [15], phosphides [16,17,18], nitrides [19,20,21] and oxides [22,23,24] have been increasingly studied, and electrocatalysts with low over-potential, high activity and high stability have been explored. Ultra-thin two-dimensional (2D) nanomaterials have unique bonding interaction methods, including single or few layers of transition metal carbon disulfide (TMD), metal oxides, etc. Strong covalent bonds extend through atoms in the plane, while weak van der Waals interactions exist between the layers. Weak interlayer bonding can easily peel these materials into thinner nanosheets containing several or single layers. These TMD ultra nanosheets usually exhibit anisotropy and have a larger surface-to-volume ratio, thereby providing high-density surface active sites [25], which is a benefit to the application of HER catalyst. So, TMD nanosheets are alternative materials [26,27,28] of platinum-based catalyst for HER. SnS_2_ is a typical two-dimensional material, which has the above-mentioned advantages of two-dimensional materials working as HER catalyst. However, the higher over-potential and Tafel slope in catalytic hydrogen evolution limit its application of HER catalyst. In our work, by doping Pt on the main carrier of SnS_2_ nanosheets, the SnS_2_-Pt composite catalyst was prepared. The SnS_2_-Pt composite catalyst shows a lower over-potential and Tafel slope, compared with SnS_2_ nanosheets. Using the synergistic effect [29,30] between SnS_2_ and platinum, the hydrogen evolution performance is effectively promoted in this work.

Herein, we designed an efficient synthesis route to prepare dispersed Pt nanoparticles anchored on ultra-thin SnS_2_ frameworks (SnS_2_–Pt). The benefit of using SnS_2_ cooperated with small amount of Pt, to help reduce the used amount of this precious metal, while keep the performance of the catalyst. The as-synthesized individual Pt nanoparticles are clearly identified through the aberration-corrected scanning transmission electron microscopy (AC-STEM). Furthermore, the ultra-thin SnS_2_ combined with Pt nanoparticles can obviously enhance the conductivity, abundant exposed active sites and efficient transfer of the HER-related electrons, which endows SnS_2_-Pt-3 with excellent HER activities in acidic electrolyte. The over-potential of the as prepared SnS_2_-Pt-3 nanocatalyst is 210 mV under the current density of 10 mA cm^−2^, while the Tafel slope of SnS_2_-Pt-3 was 126 mV dec^−1^ in the electrolyte of 0.5 M H_2_SO_4_, as well as stability. This study provides a new way to design advanced SnS_2_ catalyst with high activity, and meets the urgent needs of sustainable hydrogen economy.

## 2. Experimental Section

### 2.1. Chemicals

Tin (IV) chloride pentahydrate (SnCl_4_·5H_2_O, AR), Thioacetamide (C_2_H_5_NS), Isopropyl Alcohol (C_3_H_8_O, AR), Chloroplatinic acid hexahydrate (H_2_PtCl_6_·6H_2_O), Sulfuric acid (H_2_SO_4_, GR), ethanol absolute (CH_3_CH_2_OH, AR) were obtained from Beijing Chemical Factory, Membrane solution and Commercial 20 wt % Pt/C were provided by Shanghai Hesen Electric Co., Ltd., Shanghai, China. The chemical reagents, used in the experimental preparation, are all of analytical purity and can be used directly without further decontamination. Ultrapure water was used in this word.

### 2.2. Synthesis

In this work, 0.35 g of crystalline tin tetrachloride and 0.3 g of thioacetamide (molar ratio: 1:4) were weighed and placed in a beakers, followed by 40 mL of isopropyl alcohol, which was stirred continuously for 30 min to form a uniform and transparent solution. Then, the solution was transferred to a 50 mL Teflon lined stainless steel autoclave. After that, the autoclave was sealed and placed in an oven and reacted at 180 °C for 24 h. The reactor was cooled to room temperature and the sediments were collected. The sediments were rinsed with ethanol and deionized water for several times, and centrifuged at 60 °C for 12 h to obtain ultra-thin nanometer SnS_2_ catalyst.

The SnS_2_ powder prepared above and hexahydrate of chloroplatinic acid (1 mmol/L) were dissolved in anhydrous ethanol at a mass ratio of 50:1, stirred continuously for 3 h, centrifuged and rinsed with deionized water for several times, dried at 12 h at 60 °C and calcined in vacuum at Ar 200 °C for 2 h to obtain SnS_2_-Pt.

### 2.3. Characterization

The X-ray diffraction (XRD) pattern of SnS_2_ nanocatalyst were measured on XRD instrument (Rigaku, SmartLab 9 KW, Japan). The range of 2θ was set at 10–75° and the scanning rate was set at 10° min^−1^. The morphology of the SnS_2_ nanocatalyst were obtained on Verios 460 L, and the structural characterization of SnS_2_ nanocatalyst were carried on a High-Resolution Transmission Electron Microscope (Talos F200X, FEI, Hillsboro, OR, USA) and a Transmission Electron Microscope with A Probe Corrector (Titan G2 300, FEI, Hillsboro, OR, USA). The X-ray photoelectron spectroscopy (XPS) and binding energy were calibrated with reference to C1’s main peak of 284.8 eV.

### 2.4. Electrochemical Measurements

Altl electrochemical measurements [31,32,33,34] were performed in a typical three-electrode system on a CHI 760E electrochemical workstation at room temperature using a carbon paper (HCP030N, 0.3 mm of thick), modified with the catalysts as working electrode, an Ag/AgCl (in 0.5 M H_2_SO_4_) electrode as reference electrode, a carbon rod as the counter electrode. To prepare the catalyst ink, the catalyst (6 mg) was dissolved in anhydrous ethanol (500 microliter) and the Nafion (500 microliter) mixture of 0.5 wt % by ultrasonic dispersion for 30 min. The catalyst (200 microliter) was dropped on the surface of carbon paper (1 cm × 1 cm, with a load of 1.2 mg cm^−2^) and dried at room temperature. SnS_2_, SnS_2_-Pt (scanning rate:10 mV s^−1^) were subjected to linear scanning voltammetry (LSV) under the condition of 0.5 M H_2_SO_4_ (before HER measurement, the electrolyte was purified with pure N_2_ gas for 30 min to remove the dissolved oxygen). Cyclic voltammetry (CV) scanning rate was 100 mV s^−1^. By plotting the logarithmic current density of overpotential, the Tafel curve is obtained. Electrochemical impedance spectroscopy (EIS) was measured using the CHI 760E (Beijing, China) electrochemical workstation with an AC voltage amplitude of 5 mV and a frequency range of 0.01 Hz to 100 KHz at 0.5 M H_2_SO_4_. All data is compensated by i*R*. Electrochemical bi-layer capacitance (*C*_dl_) for a binary free process was evaluated five times using cyclic volt-ampere method at five different scanning rates (25, 30, 35, 40 and 45 mV s^−1^).

## 3. Results and Discussion

In this work, we synthesized ultra-thin SnS_2_ nanosheets by a simple hydrothermal method [35,36] (see the Experimental Section for details), and further low-concentration Pt precursor to stir and adsorb to form SnS_2_-Pt-3 materials, which were calcined to make the platinum particles and SnS_2_ combine perfectly. By testing the hydrogen evolution performance of these four catalysts, we found that the SnS_2_-Pt-3 catalyst has the best HER performance, so we focused on its structural characterization. The scanning electron microscopy (SEM) and transmission electron microscope (TEM) images are all shown phase and structure information of SnS_2_-Pt-3 catalyst, while the results XPS are used to obtain the element valence state and composition analysis of SnS_2_-Pt-3 catalyst. As shown in Figure 1, SnS_2_-Pt-3 nanosheets were prepared according to the mass ratio of SnS_2_ to H_2_PtCl_6_·H_2_O (100 wt %: 5 wt %), in order to explore the influence of different Pt doping amount on catalytic hydrogen evolution performance, for comparison, we have prepared four different Pt-doping amount samples in our work. The amount of Pt are 0.04 wt %, 0.09 wt %, 0.5 wt % and 0.8 wt %, respectively. The four samples are renamed as SnS_2_-Pt-1, SnS_2_-Pt-2, SnS_2_-Pt-3 and SnS_2_-Pt-4, respectively.

Figure 2a shows the SEM image, which is a flower-like SnS_2_ nanostructure with similar morphology to pure SnS_2_ (Appendix A). By observing the low-magnification SEM image, no obvious large Pt clusters were found in SnS_2_-Pt-3. The TEM image of SnS_2_-Pt-3 is shown in Figure 2b. It can be seen that it is a very thin nanosheet, which is also confirmed in SEM image. By observing the high-resolution TEM (HRTEM) image (Figure 2c), we can see that the interfacial spacing d = 0.311 nm on surface (100) is very similar to the pure SnS_2_ HRTEM image (Appendix A). Figure 2d,e is atomic resolution high-angle annular dark-field scanning TEM (HAADF-STEM) images [37], which show some Pt clusters represented by red circles. By comparing the selected area electron diffraction (SAED) diagrams before, and after, doping, there is no obvious change in the interplanar spacing of SnS_2_ (SnS_2_ (100) plane with lattice spacing of 0.311 nm) (Figure 2c and Appendix A). Inductively coupled plasma-atomic emission spectroscopy (ICP-MS) revealed that the Pt content of SnS_2_-Pt-3 is 0.5 wt %. In addition, the HAADF image and the corresponding energy-dispersive X-ray spectroscopy (EDS) (Figure 2f) show that Sn, S and Pt element are uniformly distributed in SnS_2_-Pt-3 nanosheets, and the corresponding element ratios are shown in Appendix A.

The X-ray photoelectron spectroscopy (XPS) was used to determine the elemental composition of SnS_2_ nanosheets doped with different proportions of Pt precursor. As shown in the high-resolution XPS spectra of pure SnS_2_ in Appendix A, the two main peaks of Sn are 3d_5/2_ of 486.9 eV and 3d_3/2_ of 495.3 eV are in line with Sn^4+^, and the two main peaks of S are 2p_3/2_ of 162.0 eV and 2p_1/2_ of 163.2 eV are assigned to S^2−^, respectively [38]. By analyzing the distribution of Sn, S elements and the ratio of Sn/S (1:2) in the EDS mapping (Appendix A) and the XRD diagram (Appendix A), indicating that suggesting a rational stoichiometric composition of SnS_2_. Besides, in the high-resolution XPS spectra of SnS_2_-Pt-3 (Figure 3a,b), the Sn are 3d_5/2_ of 486.6 eV and 3d_3/2_ of 495.0 eV, and S are 2p_3/2_ of 161.7 eV and 2p_1/2_ of 162.9 eV and in the high-resolution XPS spectra of SnS_2_-Pt-4 (Appendix A), the Sn are 3d_5/2_ of 486.7 eV and 3d_3/2_ of 495.1 eV, and S are 2p_3/2_ of 161.8 eV and 2p_1/2_ of 163.0 eV. Compared with pure SnS_2_, all the peak positions of Sn 3d and the S 2p regions in SnS_2_-Pt-3 were lower binding energy shifted. Moreover, two Pt 4f peaks at 75.66 eV (4f_5/2_) and 72.43 eV (4f_7/2_) of SnS_2_-Pt-3 (Figure 3c) and two Pt 4f peaks at 75.36 eV (4f_5/2_) and 71.98 eV (4f_7/2_) of SnS_2_-Pt-4 (Appendix A) were observed in the XPS spectrum, indicative of Pt^0^. The detectable Pt^0^ signal indicates that Pt crystal grains exist in SnS_2_-Pt-3 and SnS_2_-Pt-4, which further confirming HAADF-STEM and XRD results (Figure 2e and Appendix A). The whole XPS spectrum of pure SnS_2_, SnS_2_-Pt-3 and SnS_2_-Pt-4 is shown in Appendix A. These three samples show similar chemical composition of Sn, S, Pt, O. No obvious change can be observed.

The HER performance of SnS_2_-Pt-3 was measured by the carbon paper electrode test and compared against pure SnS_2_, SnS_2_-Pt-4 and commercial 20 wt % Pt/C in 0.5 M H_2_SO_4_ solution using a three-electrode system (see the Experimental Section for details) [39]. In order to reduce the experimental error caused by solution resistance, the initial data, obtained during the whole test process, were calibrated by ohmic potential drop unless special instructions, and all potential data obtained in this working electrochemical test were relative to reversible hydrogen electrode (RHE). The work electrode was scanned by CV several times until it reached a stable state. Figure 4a shows the LSV curve of the electrocatalytic hydrogen evolution of commercial Pt/C, pure SnS_2_, SnS_2_-Pt-3 and SnS_2_-Pt-4 at a scanning speed of 10 mV s^−1^. The cathode current of 10 mA cm^−2^ of SnS_2_-Pt-3, only a low overpotential of 210 mV is needed, which is lower than pure SnS_2_ (780 mV) and SnS_2_-Pt-4 (250 mV) and not as good as commercial Pt/C (25 mV). It also proves that commercial Pt/C is indeed one of the best HER catalysts.

The Tafel slopes of SnS_2_-Pt-3 and reference materials were linearly fitted (Figure 4b), and SnS_2_ Pt-3 Tafel slopes was measured to be 126 mV dec^−1^, which was smaller than that of pure SnS_2_ (282 mV dec^−1^) and SnS_2_-Pt-4 (153 mV dec^−1^), also indicating that the favorable electrocatalytic kinetics of SnS_2_-Pt-3. According to related reports [40,41], the clustered catalyst can fully contact the surface active sites of the SnS_2_ sample, thereby, optimizing the adsorption and release of hydrogen and promoting the role of catalytic hydrogen release. The catalyst in the form of platinum particles cannot fully contact the active sites due to its large size, which reduces the hydrogen evolution efficiency. By analyzing the TEM (HAADF-STEM) image (Appendix A), more Pt in SnS_2_ (SnS_2_-Pt-4) forms the particles in the sample, while Pt in SnS_2_-Pt-3 forms the clusters, which makes SnS_2_-Pt-3 nanosheets show excellent electrocatalytic performance in HER. The result further confirms that SnS_2_-Pt-3 is an excellent electrocatalyst for HER. In order to verify, the electrochemical active surface area (ECSA) of the SnS_2_-Pt-3 and reference materials were also evaluated by measuring electrochemical double-layer capacitance (C_dl_). Select the non-Faraday region and measure the CV curves of Pt-C, pure SnS_2_, SnS_2_- Pt-1, SnS_2_-Pt-2, SnS_2_-Pt-3 and SnS_2_-Pt-4 at scan rates of 25, 30, 35, 40 and 45 mV s^−1^ (Appendix A). The electrochemically active surface area of the catalyst was obtained by linear fitting. As shown in Figure 4c, the C_dl_ values of Pt-C, pure SnS_2_, SnS_2_-Pt-3 and SnS_2_-Pt-4 catalysts are 5.9, 1.4, 6.3, and 4.4 mF cm^−2^, respectively. The C_dl_ value of the SnS_2_-Pt-3 catalyst is higher than that of Pt-C, because the doped platinum forms platinum clusters, which can improve the conductivity of the two-dimensional SnS_2_ nanosheets, and promote faster interface charge transfer and clever electrochemistry catalysis. This confirms again that the hydrogen release efficiency of clustered platinum is better than that of platinum particles. In addition, electrochemical impedance spectroscopy (EIS) was conducted to study the influence of Pt clusters on the catalytic kinetics of electrocatalysts. As shown in Figure 4d, the charge transfer resistance (Rct) of SnS_2_-Pt-3 is much smaller than that of pure SnS_2_, indicating that the faster electron transfer and HER catalytic kinetics of SnS_2_-Pt-3. In order to evaluate the stability of SnS_2_-Pt-3 catalyst, continuous constant potential electrolysis is necessary for practical application. As shown in Figure 4e,f, the current density of SnS_2_-Pt-3 did not decrease significantly over 20 h, and the electro-catalytic stability of SnS_2_ Pt-3 could also be demonstrated after 1000 potential cycles.

By comparison with other synthesis methods (Table 1), we found that the SnS_2_ prepared by hydrothermal method is not only simple in operation, but also has enough samples in one time. The SEM and TEM images show ultra-thin morphology and structure, which are not possessed by other preparation methods. SnS_2_-Pt nanosheets were prepared by simple doping method, which greatly improved the electrocatalytic hydrogen evolution performance of SnS_2_. Compared with other methods, the electrocatalytic hydrogen evolution performance of SnS_2_ was significantly improved, which concludes that our research work is feasible and novel.

In additions, we also performed SnS_2_-Pt-1 and SnS_2_-Pt-2 HER performance (Figure 5a), and the results showed that the over-potential at 10 mA cm^−2^ current density was 290 mV and 320 mV, respectively, which were higher than SnS_2_-Pt-3. The measured Tafel slopes of SnS_2_-Pt-1 and SnS_2_-Pt-2 (Figure 5b) are 176 mV dec^−1^, and 184 mV dec^−1^, respectively. These results indicating a faster hydrogen insertion/extraction kinetics for SnS_2_-Pt-3. Meanwhile, the C_dl_ value of SnS_2_-Pt-1 and SnS_2_-Pt-2 are 3.8 mF cm^−2^ and 3.4 mF cm^−2^ (Figure 5c), also further indicating the improved electrochemically active sites of SnS_2_-Pt-3. Figure 5d shows the XRD patterns of SnS_2_-Pt-1 and SnS_2_-Pt-1, which indicates that the phase and structure information of them has no significant change compared to pure SnS_2_.

In addition, after a long-term catalytic process, the HAADF-STEM images (Appendix A), XRD (Appendix A) showed no significant changes, indicating that the morphology and structure of HAADF-STEM were well preserved. The XPS of SnS_2_-Pt-3 samples that have been tested for 20 h, the Sn are 3d_5/2_ of 487.06 eV and 3d_3/2_ of 495.49 eV, and S are 2p_3/2_ of 162.18 eV and 2p_1/2_ of 163.37 eV, two Pt 4f peaks at 76.12 eV (4f_5/2_) and 72.85 eV (4f_7/2_). The whole XPS spectrum is shown in Appendix A, no obvious change can be observed, which further confirmed its excellent structural stability.

## 4. Conclusions

In summary, ultra-thin SnS_2_ nanosheets were prepared by a simple hydrothermal synthesis method, and then small Pt clusters were anchored on ultra-thin SnS_2_ to form SnS_2_-Pt-3 nanosheets. Furthermore, compared with pure ultra-thin SnS_2_, the prepared ultra-thin SnS_2_-Pt-3 nanosheets shows obviously superior HER performance. Under the current density of 10 mA cm^−2^, its overpotential was 210 mV and its Tafel slope was 126 mV dec^−1^ in 0.5 M H_2_SO_4_ and no obvious attenuation phenomenon was observed after 20 h chronoamperometry, as well as stability. All these indicate that SnS_2_-Pt-3 nanosheets catalyst have broad application prospects in H_2_ production, energy supply and electrochemical reaction. This prominent HER performance is due to the collaborative effect between the ultra-thin nanosheet structure and Pt clusters, which together enhance the active site of the catalyst. Our work also proves that synergy is an available strategy to improve the electrocatalytic performance of two-dimensional materials.

## Figures and Tables

**Figure 1 nanomaterials-10-02337-f001:**
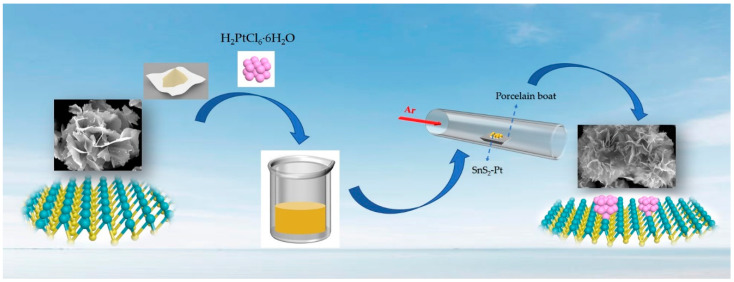
Synthesis diagram of ultra-thin SnS_2_-Pt-3 nanosheets.

**Figure 2 nanomaterials-10-02337-f002:**
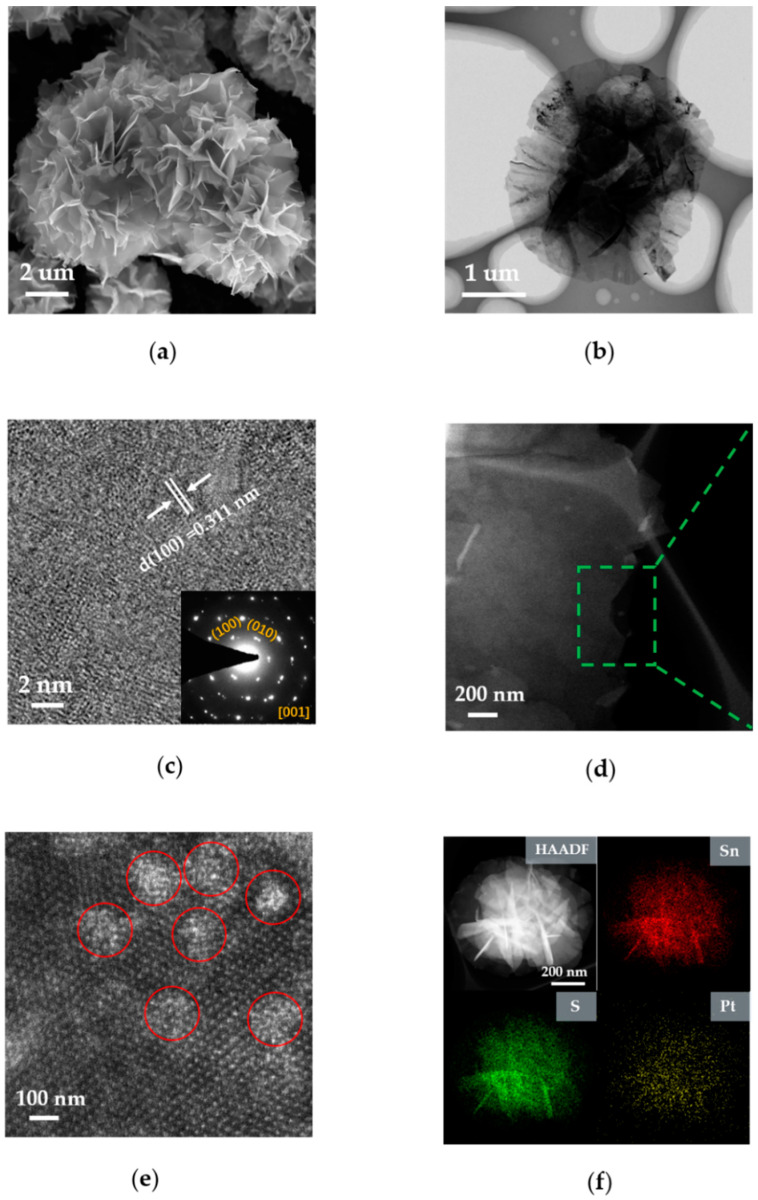
Structures of SnS_2_-Pt-3. (**a**) The SEM image of SnS_2_ nanostructure. (**b**) The TEM image of low-magnification morphology of SnS_2_-Pt-3 nanosheets. (**c**) SnS_2_-Pt-3 of high-resolution transmission diagram, in which is the corresponding SAED pattern. (**d**,**e**) Atomic resolution HAADF-STEM image with some individual Pt clusters represented by red circles. (**f**) The HAADF image and EDS mapping of Sn, S and Pt element from a SnS_2_-Pt-3 nanosheet.

**Figure 3 nanomaterials-10-02337-f003:**
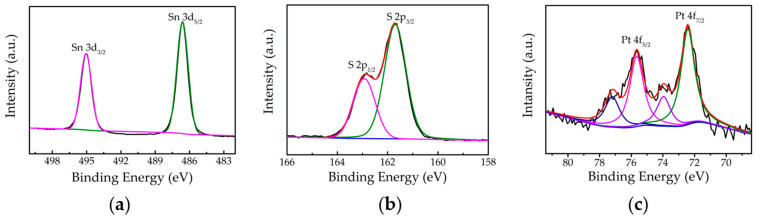
Chemical state analysis of SnS_2_-Pt-3. (**a**,**b**) are Sn 3d and S 2p high-resolution XPS spectra of SnS_2_-Pt-3. (**c**) the high-resolution XPS spectra of Pt 4f.

**Figure 4 nanomaterials-10-02337-f004:**
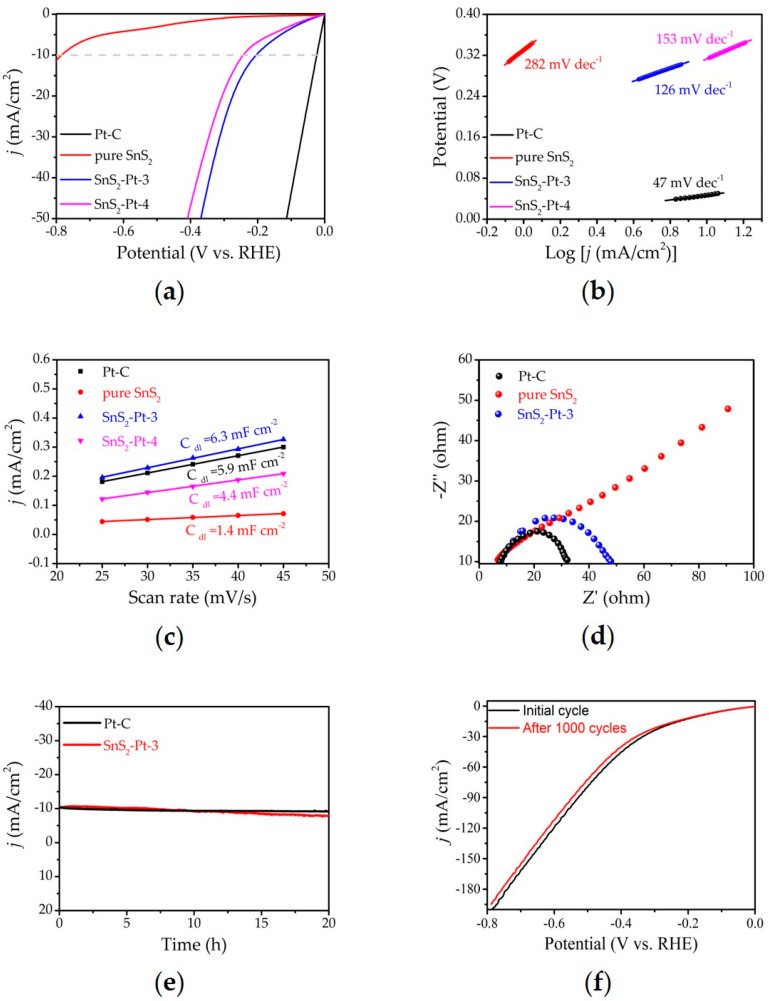
HER of the SnS_2_-Pt-3. (**a**) LSV curves of the commercial Pt/C, the pure SnS_2_ nanosheets and the SnS_2_-Pt-3 ones. (**b**) Tafel plots corresponding to (**a**). (**c**) Electrochemical active surface area with different mass ratio. (**d**) EIS measurement of pure SnS_2_ and SnS_2_-Pt-3. (**e**) I-T distribution diagram of SnS_2_-Pt-3 nanosheets. (**f**) LSV curves of the SnS_2_-Pt-3 before and after 1000 potential cycles.

**Figure 5 nanomaterials-10-02337-f005:**
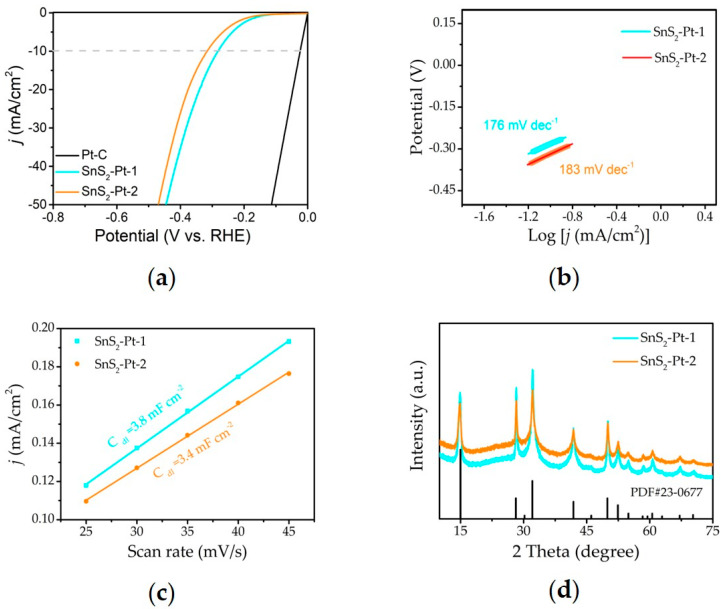
Structural and performance characterization of SnS_2_-Pt-1 and SnS_2_-Pt-2. (**a**) The LSV curves under hydrogen evolution. (**b**) The Tafel slopes. (**c**) Electrochemical active surface area. (**d**) XRD.

**Table 1 nanomaterials-10-02337-t001:** Comparison table of electrocatalytic performance data of SnS_2_ and its hybrids.

Electrode Material	Synthesis Method	Electrolyte	Overpotential at 10 mA/cm^2^	Tafel Slope	Reference
SnS_2_ SnS_2_-Pt-3	Hydrothermal synthesis	0.5 M H_2_SO_4_	−780 mV −210 mV	282 mV dec^−1^ 126 mV dec^−1^	This work
SnS_2_ MoS_2_/SnS_2_	Hydrothermal method	0.5 M H_2_SO_4_	−288 mV −580 mV	76 mV dec^−1^ 50 mV dec^−1^	Ref [42]
MoSe_2_ MoSe_2_/SnS_2_-2.5	Hydrothermal method	1.0 M KOH	−367 mV −285 mV	149 mV dec^−1^ 109 mV dec^−1^	Ref [43]
MoS_2_ MoS_2_/SnS_2_-2.5	Hydrothermal method	0.5 M H_2_SO_4_	−419 mV −343 mV	216 mV dec^−1^ 157 mV dec^−1^	Ref [43]
SnS_2_ SnS_2_/G	Solid-state ball-milling approach	1.0 M KOH	−600 mV −360 mV	375 mV dec^−1^ 257 mV dec^−1^	Ref [44]
SnS_2_ Sn_0.3_W_0.7_S_2_	Hydrothermal method	0.5 M H_2_SO_4_	−481 mV −345 mV	398 mV dec^−1^ 114 mV dec^−1^	Ref [45]
SnS_2_ 5% Mo-SnS 10% Mo-SnS	Colloidal technique	0.5 M H_2_SO_4_	−600 mV −486 mV −377 mV	328 mV dec^−1^ 177 mV dec^−1^ 100 mV dec^−1^	Ref [46]

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
