# Peer review of "Ultra-Thin SnS2-Pt Nanocatalyst for Efficient Hydrogen Evolution Reaction"

_nanomaterials, 2020, doi:10.3390/nano10122337_

Round 1
Reviewer 1 Report
The manuscript “Ultra-thin SnS2 Nanocatalyst for Efficient Hydrogen 2 Evolution Reaction“ by Y. Yu et al. reports on the synthesis of SnS2-Pt nanocomposites for HER. SEM, TEM and XPS were used to characterise the samples.
The makeup and total quality of the manuscript should be improved before it could be accepted for publication in nanomaterials.
- As reported by the authors, the main target for HER is replacing of Pt. The authors have to explain the benefit of their nanocomposite because there is still Pt as active compound. More generally, the authors have to explain the novelty of their work.
- The synthesis of SnS2 nanosheet by hydrothermal is not new, see DOI: https://doi.org/10.1166/sam.2013.1667 or Journal of Materials Science, 3, 2019.
- This sentence has to be corrected, “no obvious large Pt particles were found in pure SnS2 …There is no sense to find Pt in pure SnS2.
- The authors have to explain this sentence, “Inductively coupled plasma-atomic emission spectroscopy (ICP-AES) 125 revealed that the Pt content of SnS2-Pt is 0.5 wt%.”. There are 3 different SnS2-Pt nanocomposites with 1, 2 and 10wt% of Pt, the amount of Pt (0.5wt%) is same for all or only one was analysed?
- The authors have to correct this sentence, “The ratio of Sn/S is about 1:2, indicating that suggesting a rational stoichiometric composition of SnS2.”. XPS is only an extreme surface analysis, it is not possible to conclude about the purity without complementary techniques. Is it possible to perform XRD or Raman?
- The authors have to explain clearly which SnS2-x Pt sample was analysed. It is not clear at all. Most of the characterisation, only SnS2-Pt is reported (see SEM, TEM, XPS…). The experimental part should be improved too.
- Same comments for HER. This point should be corrected. “Fig. 4a shows the LSV curve of the electrocatalytic hydrogen evolution of commercial Pt/C, pure 158 SnS2, SnS2-x3 Pt and SnS2-Pt at a scanning speed of 10 mV s-1”. What is SnS2-Pt? x = 5wt%? If yes, maybe it is better to call the samples SnS2-x1 (1wt%), SnS2-x2 (2wt%), SnS2-x5 (5wt%) and SnS2-x10 (10wt%).
Author Response
Response to the comments of Reviewer 1:
Comments:
- As reported by the authors, the main target for HER is replacing of Pt. The authors have to explain the benefit of their nanocomposite because there is still Pt as active compound. More generally, the authors have to explain the novelty of their work. The synthesis of SnS2 nanosheet by hydrothermal is not new, see DOI: https://doi.org/10.1166/sam.2013.1667 or Journal of Materials Science, 3, 2019.
Response: Thank you very much for reviewing our manuscript. We really appreciate your helpful and constructive comments. The platinum-based catalyst has high hydrogen production efficiency and good stability in the direction of electrocatalysis, but its high price and scarcity limit its wide application in industrial production.
Ultra-thin two-dimensional nanomaterials (2D) have unique bonding interaction methods, including single or few layers of transition metal carbon disulfide (TMD), metal oxides, etc. Strong covalent bonds extend through atoms in the plane, while weak van der Waals interactions exist between the layers. Weak interlayer bonding can easily peel these materials into thinner nanosheets containing several or single layers. These TMD ultra nanosheets usually exhibit anisotropy and have a larger surface-to-volume ratio, thereby providing high-density surface active sites, which is benefit to the application of HER catalyst. So, TMD nanosheets are though of the alternative materials of platinum-based catalyst for HER.
SnS2 is a typical two-dimensional material, which has the above-mentioned advantages of two-dimensional materials working as HER catalyst. However, the higher over-potential and Tafel slope in catalytic hydrogen evolution limit its application of HER catalyst. In our work, by doping Pt on the main carrier of SnS2 nanosheets, the SnS2-Pt composite catalyst was prepared. The SnS2-Pt composite catalyst shows a lower over-potential and Tafel slope compared with SnS2 nanosheets. Using the synergistic effect between SnS2 and platinum, the hydrogen evolution performance is effectively promoted in this work.
The above discussions have been added in the revised version of our manuscript. They are presented in Pages 1 and 2 of the main text file. For your convenience, all changes resulting from the reviewers' comments have been highlighted in yellow in the main text.
Thank you for your reminding. According to your suggestion, we have added the corresponding papers as references 35-36.
- This sentence has to be corrected, “no obvious large Pt clusters were found in pure SnS2 …There is no sense to find Pt in pure SnS2. The authors have to correct this sentence, “The ratio of Sn/S is about 1:2, indicating that suggesting a rational stoichiometric composition of SnS2.”. XPS is only an extreme surface analysis, it is not possible to conclude about the purity without complementary techniques. Is it possible to perform XRD or Raman?
Response: Thank you very much for reviewing our manuscript. We really appreciate your helpful and constructive comments. The sentence “By observing the low-magnification SEM image, no obvious large Pt clusters were found in pure SnS2” has been revised to “By observing the low-magnification SEM image, no obvious large Pt clusters was found in SnS2-Pt-3.” Actually, the phase composition of SnS2 are determined by XRD, XPS and EDS. XRD is used to obtain the phase and crystal structure information of SnS2 from a macroscopic perspective. The elemental composition and chemical state on the surface of SnS2 can be obtained by analyzing the XPS spectrum. The distribution of element and the ratio of Sn/S also can be obtained by EDS mapping from a microscopic perspective. The corresponding sentence in the text “The ratio of Sn/S is about 1:2, indicating that suggesting a rational stoichiometric composition of SnS2.” has been corrected to “By analyzing the distribution of Sn, S elements and the ratio of Sn/S (1:2) in the EDS mapping (Table S2) and the XRD diagram (Fig. S2f), indicating that suggesting a rational stoichiometric composition of SnS2.”.
The above results have been changed in the revised version of our manuscript. They are presented in Pages 3 and 5 of the main text file.
- The authors have to explain this sentence, “Inductively coupled plasma-atomic emission spectroscopy (ICP-AES) 125 revealed that the Pt content of SnS2-Pt is 0.5 wt%.”. There are 3 different SnS2-Pt nanocomposites with 1, 2 and 10wt% of Pt, the amount of Pt (0.5wt%) is same for all or only one was analysed? This point should be corrected. “Fig. 4a shows the LSV curve of the electrocatalytic hydrogen evolution of commercial Pt/C, pure 158 SnS2, SnS2-x3 Pt and SnS2-Pt at a scanning speed of 10 mV s-1”. What is SnS2-Pt?
Response: Thank you for your comments. The sentence “Inductively coupled plasma-atomic emission spectroscopy (ICP-MS) 125 revealed that the Pt content of SnS2-Pt is 0.5 wt%.” in the main text has been corrected to “Inductively coupled plasma-atomic emission spectroscopy (ICP-AES) revealed that the Pt content of SnS2-Pt-3 is 0.5 wt%. We have prepared four different Pt-doping amount samples in our work. The amount of Pt are 0.04 wt%, 0.09 wt%, 0.5 wt% and 0.8 wt%, respectively. The four samples are renamed as SnS2-Pt-1, SnS2-Pt-2, SnS2-Pt-3 and SnS2-Pt-4, respectively.”
- The authors have to explain clearly which SnS2-x Pt sample was analysed. It is not clear at all. Most of the characterisation, only SnS2-Pt is reported (see SEM, TEM, XPS…). The experimental part should be improved too.
Response: Thank you very much for reviewing our manuscript. In this work, prepared four different Pt-doping amount samples in our work, named SnS2-Pt-1, SnS2-Pt-2, SnS2-Pt-3 and SnS2-Pt-4. By testing the hydrogen evolution performance of these four catalysts, we found that the SnS2-Pt-3 catalyst has the best HER performance, so we focused on its structural characterization. The SEM and TEM images are all shown phase and structure information of SnS2-Pt-3 catalyst, while the results XPS are used to obtain the element valence state and composition analysis of SnS2-Pt-3 catalyst.
The above results have been changed in the revised version of our manuscript. They are presented in Pages 3 of the main text file.
Reviewer 2 Report
In my opinion, the manuscript is well organized and written, and has interesting data that can suitable for high prestige journal Nanomaterials. But here are some things that authors should address: The synthesis of the catalyst mentioned in the manuscript has not been compared and studied with other methods of preparing this catalyst, basically in this manuscript, there is not enough reference to the novelty of the research! For the XPS spectrum, it is better to include the whole spectrum! In the experimental part, it is better to add some references such as "Journal of Molecular Liquids 215, 253-257, 2016" and "Electroanalysis 27 (7), 1766-1773, 2015" and "Current Analytical Chemistry 13 (1), 81-86, 2017" and "Journal of the Taiwan Institute of Chemical Engineers 80, 989-996, 2017" for the parts of "2.4. Electrochemical measurements" and "2.2. Synthesis".
Author Response
Response to the comments of Reviewer 2:
- The synthesis of the catalyst mentioned in the manuscript has not been compared and studied with other methods of preparing this catalyst, basically in this manuscript, there is not enough reference to the novelty of the research!
Response: Thank you for your comments. We have added the comparation of synthesis methods and the performances of HER for SnS2-based catalyst, as shown in Table S3. By comparation with other synthesis methods, we found that the SnS2 prepared by hydrothermal method is not only simple in operation, but also has enough samples in one time. Its SEM and TEM images can show ultra-thin morphology and structure, which are not possessed by other preparation methods. SnS2-Pt nanosheets were prepared by simple doping method, which greatly improved the electrocatalytic hydrogen evolution performance of SnS2. Compared with other methods, the electrocatalytic hydrogen evolution performance of SnS2 was significantly improved.
The above results have been changed in the revised version of our manuscript. They are presented in Pages 10 of the SI file.
|
Electrode material |
Synthesis method |
Electrolyte |
Overpotential at 10mA/cm2 |
Tafel slope |
References |
|
SnS2 SnS2-Pt-3 |
Hydrothermal synthesis |
0.5M H2SO4 |
-780 mV -210 mV |
282 mV dec-1 126 mV dec-1 |
This work |
|
SnS2 MoS2/SnS2 |
Hydrothermal method |
0.5M H2SO4 |
-288 mV -580 mV |
76 mV dec-1 50 mV dec-1 |
Mol. Catal., 2020, 487, 110890 |
|
MoSe2 MoSe2/SnS2-2.5 |
Hydrothermal method |
1.0 M KOH |
-367 mV -285 mV |
149 mV dec-1 109 mV dec-1 |
Nano Energy, 2019, 64, UNSP 103918 |
|
MoS2 MoS2/SnS2-2.5 |
Hydrothermal method |
0.5M H2SO4 |
-419 mV -343 mV |
216 mV dec-1 157 mV dec-1 |
Nano Energy, 2019, 64, UNSP 103918 |
|
SnS2 SnS2/G |
Solid-state ball-milling approach |
1.0 M KOH |
-600 mV -360 mV |
375 mV dec-1 257 mV dec-1 |
ACS Appl. Energy Mater., 2020, 3, 4995-5005 |
|
SnS2 Sn0.3W0.7S2 |
Hydrothermal method |
0.5M H2SO4 |
-481 mV -345 mV |
398 mV dec-1 114 mV dec-1 |
Adv. Funct. Mater., 2020, 30, 1906069 |
|
SnS2 5% Mo-SnS 10% Mo-SnS |
Colloidal technique |
0.5M H2SO4 |
-600 mV -486 mV -377 mV
|
328 mV dec-1 177 mV dec-1 100 mV dec-1 |
Chem. Eur. J., 2020, 26, 6679-6685 |
Table S3. Comparison table of electrocatalytic performance data of SnS2 and its hybrids.
- For the XPS spectrum, it is better to include the whole spectrum!
Response: Thank you very much for reviewing our manuscript. We really appreciate your helpful and constructive comments. The whole spectrum of XPS has been added as shown in Fig. S3 in the supporting information.
Figure S5. High-resolution XPS spectra. (a) Pure SnS2, SnS2-x5 Pt and SnS2-Pt-4. (b) SnS2-Pt-3 after 20-hour test.
- In the experimental part, it is better to add some references such as "Journal of Molecular Liquids 215, 253-257, 2016" and "Electroanalysis 27 (7), 1766-1773, 2015" and "Current Analytical Chemistry 13 (1), 81-86, 2017" and "Journal of the Taiwan Institute of Chemical Engineers 80, 989-996, 2017" for the parts of "2.4. Electrochemical measurements" and "2.2. Synthesis".
Thank you for your reminding. According to your suggestion, we have added the corresponding papers as references 31-34.

Reviewer 3 Report
The manuscript entitled „Ultra-thin SnS2 Nanocatalyst for Efficient Hydrogen Evolution Reaction“ has been submitted to be considered for publication on Nanomaterials. The authors have performed synthesis and thorough characterization of the catalyst. However, to warrant a better understanding for readers, the manuscript should be revised according to comments below:
1.Title: The authors mentioned only SnS2 in the title, which may make the readers (at least I am) at first think that it contains only SnS2 and be able to show efficient HER due to its nanostructure. Thus, please name it SnS2-Pt, just as stated in abstract and throughout the manuscript.
2.Introduction:
The authors introduced the TMDs as alternative for Pt, but again came up with the idea of using Pt in SnS2-Pt. This made the discussion pointless, so please mention the benefit of using SnS2 cooperated with “small amount” of Pt, to help reduce the used amount of this precious metal, while trying to keep the performance of the catalyst.
Line 41: Explain deeper why TMDs material is considered a good potential HER catalyst for its special CdI2-type layered structure? What made this structure special?
The authors should included at least a short introduction/survey about previous works on the same materials, discussion what is new/different in this work as compared to others. For example: Paper “Solution-Processed Ultrathin SnS2–Pt Nanoplates for Photoelectrochemical Water Oxidation”, from ACS Appl. Mater. Interfaces 2019, 11, 7, 6918–6926. Or the paper entitled „Efficiently Synergistic Hydrogen Evolution Realized by Trace Amount of Pt-Decorated Defect-Rich SnS2 Nanosheets“ from ACS Appl. Mater. Interfaces 2017, 9, 43, 37750–37759. Or „Facile Solvothermal Synthesis of Hybrid SnS2/Platinum Nanoparticles for Hydrogen Peroxide Biosensing“ from Journal of Bionanoscience, Volume 9, Number 5….
3.Experimental Section:
Line 71: “autocloner”?? does it mean “autoclave” ?
Line 82: What is Verios 460L? How can the authors obtained morphology of the SnS2 nanocatalyst on Verios 460L. Please describe the real method name, and write down the company, country, etc… of the machine.
Line 78, 95, please be consistent: Pt-SnS2 -> SnS2-Pt
4.Results and Discussion
Line 108 and 111-112: The authors had SnS2-Pt (mass ratio SnS2/H2PtCl6.H2O is 100:5) and 3 other types of SnS2-Pt with different amount of Pt: x1, x2, x3. -> Please explain how they calculate this x1, x2, x3? Wt% of what?
Page 5: I think it is important to put the data about the effect of loading in this main manuscript, not in the SI, since it makes the discussion really difficult to follow. Thus, put the data of Fig. S4 here. Explain why more Pt in SnS2 induced higher over-potential? This is quite illogical and need to be explained in details. It may be supportive to give the SEM/TEM images of these higher Pt loading samples.
For the electrochemical active surface area, please provide the whole CVs for all the tested materials. According to ASTM standard, CV test should be performed at very low scan rate, not 100 mV/s or 20 mV/s; but as low as 0.1667 mV/s. Faster scan rates often result in distorted data, because the sample cannot remain relatively stable.
Stability test should be performed longer than 20h…
Last but not least, the meaning of synthesizing SnS2 with Pt is to reduce Pt loading, please calculate: how much % of Pt was converted/produced from the acid H2PtCl6? (to see the reaction efficiency); then how much performance was lost compared to the same amount of Pt in Pt-C commercial catalyst. Then come to conclusion, how successful of this Pt saving is.
Author Response
Response to the comments of Reviewer 3:
Comments:
- Title: The authors mentioned only SnS2 in the title, which may make the readers (at least I am) at first think that it contains only SnS2 and be able to show efficient HER due to its nanostructure. Thus, please name it SnS2-Pt, just as stated in abstract and throughout the manuscript.
Response: Thank you very much for reviewing our manuscript. The title has been revised to “Ultra-thin SnS2-Pt Nanocatalyst for Efficient Hydrogen Evolution Reaction.”
- Introduction:
The authors introduced the TMDs as alternative for Pt, but again came up with the idea of using Pt in SnS2-Pt. This made the discussion pointless, so please mention the benefit of using SnS2 cooperated with “small amount” of Pt, to help reduce the used amount of this precious metal, while trying to keep the performance of the catalyst.
Line 41: Explain deeper why TMDs material is considered a good potential HER catalyst for its special CdI2-type layered structure? What made this structure special?
The authors should included at least a short introduction/survey about previous works on the same materials, discussion what is new/different in this work as compared to others.
Response: Thank you for your comments. We have added “the benefit of using SnS2 cooperated with small amount of Pt, to help reduce the used amount of this precious metal, while keep the performance of the catalyst.” in the section of introduction.
HER is the cathodic reaction in the electrolysis of water whereby the half-reaction is
2H+ + 2e− → H2.
In acidic electrolytes, HER occurs in two steps. Commencing with proton adsorption by the Volmer mechanism: H+ + e– + M* → M–H (where M* denotes an adsorption site on the catalyst), the desorption of hydrogen gas then proceeds by the Heyrovsky mechanism: M–H + H+ + e– → H2 + M*, or the Tafel mechanism: 2M–H → H2 + M* [ Nat. Catal. 2018, 1, 909-921]. Therefore, having a high density of surface active sites contributes to the improvement of hydrogen production efficiency.
Ultra-thin two-dimensional nanomaterials (2D) have unique bonding interaction methods, including single or few layers of transition metal carbon disulfide (TMD), metal oxides, etc. Strong covalent bonds extend through atoms in the plane, while weak van der Waals interactions exist between the layers. Weak interlayer bonding can easily peel these materials into thinner nanosheets containing several or single layers. These TMD ultra nanosheets usually exhibit anisotropy and have a larger surface-to-volume ratio, thereby providing high-density surface active sites, which is benefit to the application of HER catalyst. So, TMD ultra nanosheets are thought of the alternative materials of platinum-based catalyst for HER.
We compared the electrocatalytic performance of SnS2 and SnS2-Pt with the electrocatalytic performance of SnS2 hybrids in other reference articles, as shown in Table S3. We found that the SnS2 prepared by hydrothermal method is not only simple in operation, but also has enough samples in one time. Its SEM and TEM images can show ultra-thin morphology and structure, which are not possessed by other preparation methods. SnS2-Pt nanosheets were prepared by simple doping method, which greatly improved the electrocatalytic hydrogen evolution performance of SnS2. Compared with other methods, the electrocatalytic hydrogen evolution performance of SnS2 was significantly improved.
|
Electrode material |
Synthesis method |
Electrolyte |
Overpotential at 10mA/cm2 |
Tafel slope |
References |
|
SnS2 SnS2-Pt-3 |
Hydrothermal synthesis |
0.5M H2SO4 |
-780 mV -210 mV |
282 mV dec-1 126 mV dec-1 |
This work |
|
SnS2 MoS2/SnS2 |
Hydrothermal method |
0.5M H2SO4 |
-288 mV -580 mV |
76 mV dec-1 50 mV dec-1 |
Mol. Catal., 2020, 487, 110890 |
|
MoSe2 MoSe2/SnS2-2.5 |
Hydrothermal method |
1.0 M KOH |
-367 mV -285 mV |
149 mV dec-1 109 mV dec-1 |
Nano Energy, 2019, 64, UNSP 103918 |
|
MoS2 MoS2/SnS2-2.5 |
Hydrothermal method |
0.5M H2SO4 |
-419 mV -343 mV |
216 mV dec-1 157 mV dec-1 |
Nano Energy, 2019, 64, UNSP 103918 |
|
SnS2 SnS2/G |
Solid-state ball-milling approach |
1.0 M KOH |
-600 mV -360 mV |
375 mV dec-1 257 mV dec-1 |
ACS Appl. Energy Mater., 2020, 3, 4995-5005 |
|
SnS2 Sn0.3W0.7S2 |
Hydrothermal method |
0.5M H2SO4 |
-481 mV -345 mV |
398 mV dec-1 114 mV dec-1 |
Adv. Funct. Mater., 2020, 30, 1906069 |
|
SnS2 5% Mo-SnS 10% Mo-SnS |
Colloidal technique |
0.5M H2SO4 |
-600 mV -486 mV -377 mV
|
328 mV dec-1 177 mV dec-1 100 mV dec-1 |
Chem. Eur. J., 2020, 26, 6679-6685 |
Table S3. Comparison table of electrocatalytic performance data of SnS2 and its hybrids.
3.Experimental Section:
Line 71: “autocloner”?? does it mean “autoclave” ?
Line 82: What is Verios 460L? How can the authors obtained morphology of the SnS2 nanocatalyst on Verios 460L. Please describe the real method name, and write down the company, country, etc… of the machine.
Line 78, 95, please be consistent: Pt-SnS2 -> SnS2-Pt
Response: Thank you for your comments. The “autocloner” in line 75 is corrected to “autoclave”. Verios 460L is the type of an ultrahigh-Resolution Scanning Electron Microscope with FEG. The low-magnification morphology information of SnS2-Pt nanosheets can be obtained by using ultrahigh-Resolution Scanning Electron Microscope.
We have corrected “Pt-SnS2” to “SnS2-Pt” in Line 82 and 99.
4.Results and Discussion
Line 108 and 111-112: The authors had SnS2-Pt (mass ratio SnS2/H2PtCl6·H2O is 100:5) and 3 other types of SnS2-Pt with different amount of Pt: x1, x2, x3. -> Please explain how they calculate this x1, x2, x3? Wt% of what?
Page 5: I think it is important to put the data about the effect of loading in this main manuscript, not in the SI, since it makes the discussion really difficult to follow. Thus, put the data of Fig. S4 here. Explain why more Pt in SnS2 induced higher over-potential? This is quite illogical and need to be explained in details. It may be supportive to give the SEM/TEM images of these higher Pt loading samples.
For the electrochemical active surface area, please provide the whole CVs for all the tested materials. According to ASTM standard, CV test should be performed at very low scan rate, not 100 mV/s or 20 mV/s; but as low as 0.1667 mV/s. Faster scan rates often result in distorted data, because the sample cannot remain relatively stable.
Stability test should be performed longer than 20h…
Last but not least, the meaning of synthesizing SnS2 with Pt is to reduce Pt loading, please calculate: how much % of Pt was converted/produced from the acid H2PtCl6? (to see the reaction efficiency); then how much performance was lost compared to the same amount of Pt in Pt-C commercial catalyst. Then come to conclusion, how successful of this Pt saving is.
Response: Thank you for your comments. In this experiment, We have prepared four different Pt-doping amount samples in our work. The amount of Pt are 0.04 wt%, 0.09 wt%, 0.5 wt% and 0.8 wt%, respectively. The four samples are renamed as SnS2-Pt-1, SnS2-Pt-2, SnS2-Pt-3 and SnS2-Pt-4, respectively. The Pt contents were calculated by the following formula,
in which, c is the content of platinum concentration measured by XPS, the unit is ppb; V is the volume of SnS2-Pt sample solution, the unit is mL; m is the weight of the weighted SnS2-Pt, the unit is mg.
Fig. 5 is placed on the main text to structural and performance characterization of SnS2-Pt-1 and SnS2-Pt-2
According to related reports, the clustered catalyst can fully contact the surface active sites of the SnS2 sample, thereby optimizing the adsorption and release of hydrogen and promoting the role of catalytic hydrogen release. The catalyst in the form of platinum particles cannot fully contact the active sites due to its large size, which reduces the hydrogen evolution efficiency[1,2]. By analyzing the TEM (HAADF-STEM) image (pictured), more Pt in SnS2 (SnS2-Pt-4) forms the particles in the sample, while Pt in SnS2-Pt-3 forms the clusters. The SEM and TEM images of higher Pt loading samples are added in Fig. S4 in Pages 7 of the SI file.
Figure S4. (a) SEM and (b) TEM of SnS2-Pt-4.
Double-layer capacitance measurement. Select the non-Faraday region and measure the CV curves of pure SnS2, SnS2-Pt-1, SnS2-Pt-2, SnS2-Pt-3 and SnS2-Pt-4 at scan rates of 25, 30, 35, 40, and 45 mV s-1(Fig. S6). The electrochemically active surface area of the catalyst was obtained by linear fitting. In order to evaluate the stability of the SnS2-Pt-3 catalyst, about 22h I-T test was performed on SnS2-Pt-3. We intercepted 20h of data. The results showed that the current density of SnS2-Pt-3 did not change over time.
Figure S5. Double-layer capacitance measurements.
Measure 500uL from the configured chloroplatinic acid with a concentration of 1mol/L, calculate m (H2PtCl6·6H2O) =0.26 mg according to the mass formula, and calculate the mass of platinum in chloroplatinic acid m1 (Pt) =98 ug. The platinum content in a 1 mg SnS2-Pt-3 sample measured by ICP is 0.5 wt%, and then m2 (Pt)=50 ug. The conversion rate of Pt in acidic H2PtCl6·6H2O is calculated to reach 51%.
References
- Zhou, M.; Bao, S.; Bard, Allen J. Probing Size and Substrate Effects on the Hydrogen Evolution Reaction by Single Isolated Pt Atoms, Atomic Clusters, and Nanoparticles. J. Am. Chem. Soc. 2019, 141, 7327–7332.
- Cheng, X.; Li, Y.; Zheng, L.; Yan, Y.; Zhang, Y.; Chen, G.; Sun, S.; Zhang, J. Highly active, stable oxidized platinum clusters as electrocatalysts for the hydrogen evolution reaction. Energy Environ. Sci. 2017, 10, 2450–2458.

Round 2
Reviewer 1 Report
The authors didn't fully reply to comment N°1 but they improved the manuscript for other points. The article could be accepted.
Author Response
Thanks very much for the reviewer 1's positive comment.
Reviewer 3 Report
Dear authors,
thank you for your revision. The manuscript is ready to be published after minor revision according to comments below:
- Please move the table S3 and the discussion into the main manuscript. This help readers have a deeper look at the available catalysts. And it is important.
- The authors wrote in the answer: The catalyst in the form of platinum particles cannot fully contact the active sites due to its large size, which reduces the hydrogen evolution efficiency. -> So then, from the active surface area and the actual Pt content in the catalyst, please state how big the different between active surface area between Pt-C and SnS2-Pt.
- Please add in the Fig. 4c, d and e, the curve for Pt-C as reference.
